

# Background independent tensor networks

**Chris Akers[1⋆] and Annie Y. Wei[2†]**

**1** Institute for Advanced Study, 1 Einstein Dr, Princeton, NJ 08540 USA
**2** Center for Theoretical Physics, Massachusetts Institute of Technology,
Cambridge, MA 02139, USA

⋆ cakers@ias.edu , † anniewei@mit.edu

## Abstract

Conventional holographic tensor networks can be described as toy holographic maps constructed from many small linear maps acting in a spatially local way, all connected together with "background entanglement", i.e. links of a fixed state, often the maximally entangled state. However, these constructions fall short of modeling real holographic maps. One reason is that their "areas" are trivial, taking the same value for all states, unlike in gravity where the geometry is dynamical. Recently, new constructions have ameliorated this issue by adding degrees of freedom that "live on the links". This makes areas non-trivial, equal to the background entanglement piece plus a new positive piece that depends on the state of the link degrees of freedom. Nevertheless, this still has the downside that there is background entanglement, and hence it only models relatively limited code subspaces in which every area has a definite minimum value. In this note, we simply point out that a version of these constructions goes one step further: they can be *background independent*, with no background entanglement in the holographic map. This is advantageous because it allows tensor networks to model holographic maps for larger code subspaces. In addition to pointing this out, we address some subtleties involved in making it work.

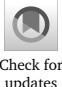

# 1 Introduction

We would like the explicit form of some holographic map. Already, what we have understood about their general structure has led to deep conceptual lessons, on topics such as the emergence of spacetime [1] and the black hole information paradox [2].

Holographic tensor networks [3–5] are a useful tool in this endeavor. Described in detail in later sections, these tensor networks can be understood as holographic maps for toy models, involving finite dimensional quantum systems. What is nice is that they share some of the striking features of real holography – such as a rudimentary version of an emergent geometry with an extra spatial dimension, and something like the quantum extremal surface (QES) formula [6–8]. Moreover, their simplicity makes it tractable to prove precise statements, for example offering rigorous insight into why a QES-like formula is inevitable in holography [3, 9, 10].

However, so far these toy holographic systems are unlike real holography in important ways. Hence the insight they can offer is limited. Two of the most glaring shortcomings are that these models do not include time evolution[1] and typically have rigid, fixed geometries. These problems are related: in gravity, the geometry is dynamical.

It seems worth thinking about whether these shortcomings can be improved, so that tensor networks might continue to offer insight into the structure of holographic maps. The goal of this note is to take a step in this direction. We will not add time evolution, but we will construct a tensor network free from any fixed geometry in a way that seems conducive to later adding time evolution resembling gravity.

Let us summarize the network now. First, how might we make a tensor network without a rigid, fixed background? As a first guess, we might imagine a model in which we consider more than just one tensor network, allowing (somehow) for a "superposition of tensor networks", each with a different geometry. This is a decent first step towards modeling gravity. However, it fails to model the fact that in gravity we do not allow arbitrary quantum states on a given geometry. There are constraints that the geometry must satisfy. These constraints are important and related to having good time evolution matching that of the dual theory. Really, we would like to construct a model akin to a "superposition of tensor networks" but in which the geometries of the tensor networks are required to satisfy some constraints.

A version of this has already been accomplished in [15–17].[2] As reviewed in Section 3, these models add degrees of freedom that "live on the links". This largely fixes the problem, because different states of these link degrees of freedom can be interpreted as different geometries, and the link degrees of freedom can be made to satisfy constraints (such as Gauss' law). Therefore these are indeed holographic tensor networks with multiple geometries obeying constraints.

That said, these existing constructions arguably only go part of the way. They involve a background geometry, in a manner we'll explain. The main goal of this note is to point out that a version of these constructions goes all the way, continuing to work even without the background geometry, once we have incorporated link degrees of freedom in the appropriate way. We explain this in Section 4. Thus we arrive at our goal: a tensor network incorporating multiple geometries, possibly all very distinct, and satisfying certain constraints. In Section 4.2 we address a subtlety in how to get this to work in one spatial dimension.

---

[1]See e.g. [11, 12] for discussions of the difficulty of including it; c.f. [13, 14] for one approach to a solution.

[2]See [18] for another idea to obtain a non-fixed geometry. Also see [19] for a model closely related to the one in this paper. The difference between that paper and this one is in which aspects of the model we study. We focus more on deriving a holographic entropy formula and related subtleties.

## 2 Conventional tensor networks

Conventional holographic tensor networks are constructed as follows [3–5]. First we define "bulk" and "boundary" Hilbert spaces, which starts by picking a graph $\Gamma$, composed of vertices and links connecting pairs of vertices. We then select some of the univalent vertices (connected to a single link) to be in the set of "boundary vertices" denoted $\{x_{\text{bdry}}\}$ and the rest to be in the set of "bulk vertices" denoted $\{x_{\text{bulk}}\}$. To each bulk vertex (respectively boundary vertex), we assign the Hilbert space of a qudit of dimension $d_{\text{bulk}}$ (respectively $d_{\text{bdry}}$). Then the total bulk Hilbert space is

$$\mathcal{H}_{\text{bulk}} = \bigotimes_{x \in \{x_{\text{bulk}}\}} \mathcal{H}_x, \tag{1}$$

and the total boundary Hilbert space is

$$\mathcal{H}_{\text{bdry}} = \bigotimes_{x \in \{x_{\text{bdry}}\}} \mathcal{H}_x. \tag{2}$$

As a definite example, consider this graph and vertex assignment:

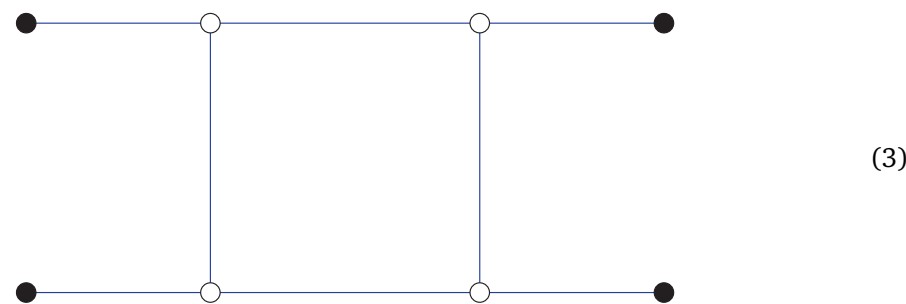

$$\tag{3}$$

The white (black) circles denote bulk (boundary) vertices. The thin blue links are drawn just to represent $\Gamma$; they have no Hilbert space associated to them.

Now we want to define a "holographic map", which is just an approximate isometry[3] $V : \mathcal{H}_{\text{bulk}} \to \mathcal{H}_{\text{bdry}}$. Holographic tensor networks are a special kind of holographic map whose typical construction proceeds as follows. To each link $(xy)$ connecting vertices $x$ and $y$, associate a bipartite Hilbert space $\mathcal{H}_{xy} \otimes \mathcal{H}_{yx}$, with $\dim \mathcal{H}_{xy} = \dim \mathcal{H}_{yx} = D_{xy}$. Now, for each link pick a state $|\phi_{xy}\rangle \in \mathcal{H}_{xy} \otimes \mathcal{H}_{yx}$. Often one chooses $|\phi_{xy}\rangle = |\text{MAX}\rangle = \sum_i |i\rangle_{xy} |i\rangle_{yx} / \sqrt{D_{xy}}$ for all links. This defines a special state on all the links that will be important momentarily,

$$\bigotimes_{\langle xy \rangle} |\phi_{xy}\rangle \in \bigotimes_{\langle xy \rangle} \mathcal{H}_{xy} \otimes \mathcal{H}_{yx}. \tag{4}$$

We have used $\langle xy \rangle$ to denote the set of all links $(xy)$. Finally we introduce the tensors. For each vertex of $\Gamma$ we associate the following collection of Hilbert spaces:

$$\mathcal{H}_{T_x} := \mathcal{H}_x \otimes \left( \bigotimes_{y \text{ nn } x} \mathcal{H}_{xy} \right), \tag{5}$$

where "$y$ nn $x$" is the set of all vertices $y$ connected to vertex $x$ by a link ($y$ "nearest neighbor" to $x$). For each *bulk* vertex, we pick a state $\langle T_x |$ in the dual Hilbert space $\mathcal{H}_{T_x}^*$. We call $\langle T_x |$ a tensor. This gives us the state

$$\bigotimes_{x \in \{x_{\text{bulk}}\}} \langle T_x | \in \bigotimes_{x \in \{x_{\text{bulk}}\}} \mathcal{H}_{T_x}^*. \tag{6}$$

---

[3]An $\varepsilon$-approximate isometry $V$ satisfies $\|V^\dagger V - \mathbb{1}\| \leq \varepsilon$ for $\varepsilon > 0$, where $\|X\|$ is the operator norm of $X$. This definition of holographic map can be generalized to allow non-isometric $V$ [2], but this subtlety won't be important for us.

We treat the boundary vertices differently.[4] Because we chose them to each have exactly one associated link, it follows that for every $x \in \{x_{\mathrm{bdry}}\}$ we have $\mathcal{H}_{T_x} = \mathcal{H}_x \otimes \mathcal{H}_{xy}$. We now assume that $d_{\mathrm{bdry}} \geq D_{xy}$ for all $xy$ and let $W_x : \mathcal{H}_{xy} \to \mathcal{H}_x$ be some isometry, possibly different for each $x$. Let $W := \prod_{x \in \{x_{\mathrm{bdry}}\}} W_x$.

We can finally define the holographic tensor network: it is the linear map $V : \mathcal{H}_{\mathrm{bulk}} \to \mathcal{H}_{\mathrm{bdry}}$ given by

$$V = W \left( \bigotimes_{x \in \{x_{\mathrm{bulk}}\}} \langle T_x | \right) \left( \bigotimes_{\langle xy \rangle} | \phi_{xy} \rangle \right). \tag{7}$$

We visualize this map for the example above as

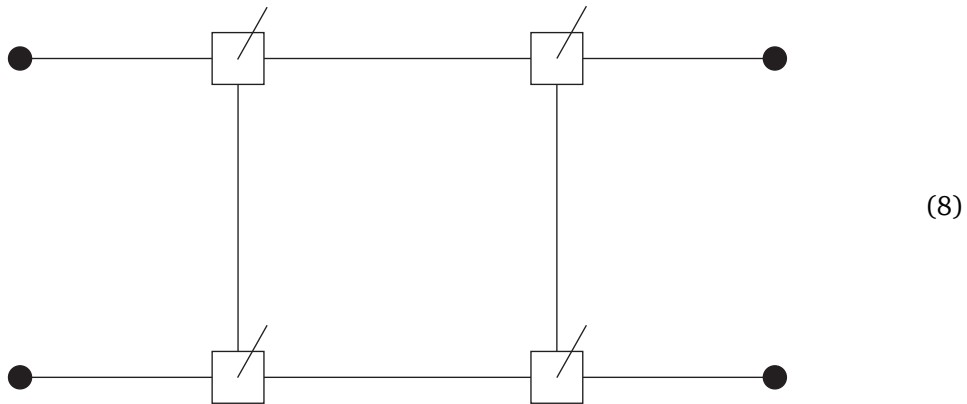

$$(8)$$

Here the white squares denote the tensors $\langle T_x |$, the black dots denote[5] $W_x$, the (vertical and horizontal) black lines connecting them denote the states $| \phi_{xy} \rangle$, and the (diagonal) black lines dangling from the tensors denote bulk inputs. Given some state $| \psi \rangle \in \mathcal{H}_{\mathrm{bulk}}$, we draw $V | \psi \rangle$ as

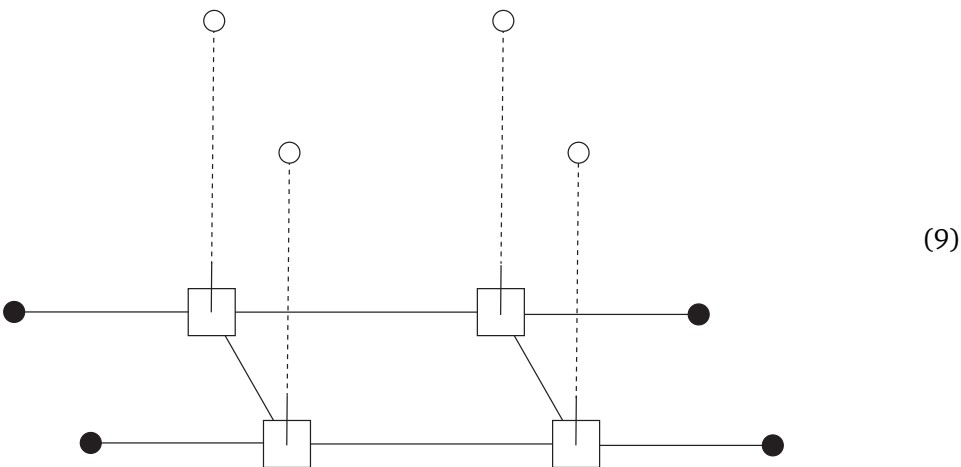

$$(9)$$

Again the white circles each denote a bulk $\mathcal{H}_x$, and the dashed lines connecting them to a bulk input denote acting that tensor on that $\mathcal{H}_x$. This completes the construction of a conventional holographic tensor network.

How do we choose each $\langle T_x |$? Different answers to this question are different tensor networks. One particularly nice option is the "random tensor network", in which we choose each independently at random [5]: pick some fiducial state $\langle 0 | \in \mathcal{H}_{T_x}^*$, and then choose a unitary $U_x$ at random according to the Haar measure on the group of unitaries acting on $\mathcal{H}_{T_x}^*$, and

---

[4]This is slightly more general than the usual constructions, which often just take $\mathcal{H}_{\mathrm{bdry}}$ to be $\bigotimes_{x \in \{x_{\mathrm{bdry}}\}} \mathcal{H}_{xy}$. We can reduce to that case by setting $W = \mathbb{1}$.

[5]We abuse notation by drawing $W_x$ and $\mathcal{H}_x$ for $x \in \{x_{\mathrm{bdry}}\}$ the same way.

let $\langle T_x | = \langle 0 | U_x$. The resulting tensor network is nice because it has a number of properties resembling the AdS/CFT duality. For example, in the regime in which $D_{xy} \gg d_{\text{bulk}}$ (for all $xy$), boundary entropies satisfy a quantum minimal surface formula. This means the following. Let $\mathcal{H}_R$ be an arbitrary auxiliary Hilbert space. Say we are given a $|\psi\rangle \in \mathcal{H}_{\text{bulk}} \otimes \mathcal{H}_R$ and resulting boundary state $V|\psi\rangle \in \mathcal{H}_{\text{bdry}} \otimes \mathcal{H}_R$. Let $B \subseteq \{x_{\text{bdry}}\}$ be a subset of the boundary vertices. The normalized density matrix of $B$ in state $V|\psi\rangle$ is then

$$\rho = \frac{\text{tr}_{\overline{B}R}[V|\psi\rangle\langle\psi|V^\dagger]}{\langle\psi|V^\dagger V|\psi\rangle}, \tag{10}$$

where $\overline{B}$ denotes the complement of $B$ in $\{x_{\text{bdry}}\}$. The von Neumann entropy of $B$ in state $V|\psi\rangle$ is defined as

$$S(B)_{V|\psi\rangle} := -\text{tr}[\rho \log \rho]. \tag{11}$$

The quantum minimal surface formula satisfied by random tensor networks says that[6]

$$S(B)_{V|\psi\rangle} = \min_b \left( \langle\psi|\hat{A}(b)|\psi\rangle + S(b)_{|\psi\rangle} \right), \tag{12}$$

where the minimization is over all homology regions $b$ of $B$.[7] Here the *area operator* is

$$\hat{A}(b) := \sum_{\langle xy\rangle \in \partial b} S(xy)_{|\phi_{xy}\rangle} \mathbb{1}, \tag{13}$$

where $\mathbb{1}$ is the identity operator on $\mathcal{H}_{\text{bulk}}$ and $\partial b$ denotes the set of links connecting $x \in b$ to $y \in \bar{b}$. (In the common case where all $|\phi_{xy}\rangle = |\text{MAX}\rangle$, it follows that $\hat{A}(b) = \sum_{xy \in \partial b} \log D_{xy} \mathbb{1}$.) The analogy to gravity comes from interpreting $\hat{A}(b)$ as the operator measuring the area associated to the quantum extremal surface. This formula then bears a strong resemblance to the quantum extremal surface (QES) formula in gravity [6–8, 20, 21].

## 3   Link degrees of freedom

One could raise the following complaint with the tensor networks from Section 2: the "area operators" $\hat{A}(b)$ are trivial: for a fixed $b$, every state is an eigenstate with the same eigenvalue. This is not like gravity in which the geometry is dynamical and areas fluctuate. To ameliorate this issue, a number of papers [15–17] have modified this setup with two tweaks. First, they add to $\mathcal{H}_{\text{bulk}}$ some degrees of freedom that can be envisioned as living on the links, such as a bulk gauge field. Second, they add to the holographic map (the tensor network) some rule for how these new link degrees of freedom should be acted on by the tensors. The effect is to modify the area operator to include a new positive term that depends non-trivially on the state of these link degrees of freedom. We present one way to do this now.[8]

The bulk Hilbert space will be that of a lattice gauge theory, along with some matter at the vertices. Let $G$ be a compact Lie group (though finite groups also work), and let $\mathcal{H}_G := L^2(G)$

---

[6]This should really be an *approximate* equality, with corrections suppressed by the ratio of various entropies in the problem. The approximation becomes increasingly good in the limit that all $D_{xy}$ are much bigger than the dimension of $\mathcal{H}_{\text{bulk}}$, and if the area of the minimal cut is much less than the area of all other cuts. From now on we will ignore these corrections. Also note that the simplest thing to compute in a random tensor network is the Renyi entropy $S_n$, which gives the formula (12) for the von Neumann entropy from analytic continuation to $n = 1$.

[7]That is, we consider all cuts $\gamma$ in $\Gamma$ that are homologous to $B$, denoting the vertices between $\gamma$ and $B$ by $b$. The minimization is over such $\gamma$.

[8]The papers [15–17] present somewhat different constructions than ours below, but they end up with the same principal result (30) and (31).

be the Hilbert space of a particle on that group manifold. This Hilbert space admits the following decomposition (see for example Appendix A of [22]):

$$\mathcal{H}_G = \bigoplus_\mu \mathcal{H}_\mu \otimes \mathcal{H}_{\mu^*}, \tag{14}$$

where $\mu$ indexes the irreducible representations (irreps) of $G$, $\mathcal{H}_\mu$ is the Hilbert space transforming under irrep $\mu$, while $\mathcal{H}_{\mu^*}$ transforms under the conjugate representation. Both have finite dimension we'll call $d_\mu$. We can therefore write an orthonormal basis as

$$|\mu; ij\rangle = |\mu; i\rangle \otimes |\mu; j\rangle \in \mathcal{H}_G, \tag{15}$$

where $\mu$ indexes the irrep and $i, j$ index the states in $\mathcal{H}_\mu, \mathcal{H}_{\mu^*}$ respectively. We will use many copies of this $\mathcal{H}_G$ momentarily. In addition, we will introduce a qudit at every vertex, with Hilbert space $\mathcal{H}_x = \mathbb{C}^{d_{\mathrm{bulk}}}$. For simplicity we will have these qudits transform trivially with respect to $G$.[9]

To build our lattice gauge theory, we take $\Gamma$ and assign an orientation to each link.[10] Then to each link we assign a Hilbert space $\mathcal{H}_G$, and to each bulk vertex we assign a Hilbert space $\mathcal{H}_x$. This is our "pre-gauged" Hilbert space $\mathcal{H}_{\mathrm{pre}}$. Our "physical" Hilbert space is the subspace $\mathcal{H}_{\mathrm{phys}} \subseteq \mathcal{H}_{\mathrm{pre}}$ that satisfies Gauss' law. A state satisfies Gauss' law if it is invariant under a gauge transformation at each vertex. In the basis (15), this means that at each bulk vertex we demand that the irreps fuse to the identity irrep.

Let's look at this in our example (3) for, say, the top left bulk vertex. There are three links attached to this vertex, and so in the pre-gauged Hilbert space a complete basis is given by states of the form

$$\mathcal{H}_{\mathrm{pre}} \text{ basis:} \tag{16}$$

Once we impose Gauss' law at this vertex, one index from each of the kets is completely determined by the Clebsch-Gordan coefficients $C^{\mu\mu'\mu''}_{ji'j''}$, which ensure that those irreps are fusing to the trivial irrep at that vertex.[11] Which of the two indices is involved depends on the orientation. We will adopt the convention that for a link oriented towards a vertex it is the second index that is involved, so

$$\mathcal{H}_{\mathrm{phys}} \text{ basis:} \quad \sum_{j,i',j''} C^{\mu\mu'\mu''}_{ji'j''} \tag{17}$$

Furthermore, $\mu, \mu'$, and $\mu''$ are related. For example if $G = SU(2)$ and $\mu = \mu' = 1$, then $\mu''$ can only be 0, 1, or 2. No other choices can fuse to the trivial irrep. Once we impose Gauss's

---

[9]We can of course allow this matter to be charged. This ends up allowing for an interesting interplay between the state of the bulk matter and the geometry. We simply start with uncharged matter for simplicity. We discuss charged matter further in Section 4.2.

[10]Different choices of orientation will end up with the same physics.

[11]For $n$-valent vertices with $n > 3$, there are generally many ways to fuse a given set of $\mu$ to the trivial irrep. That wouldn't change anything about the discussion below, but if we wanted we could simplify by always choosing the graph to be composed only of trivalent vertices.

law at every bulk vertex we arrive at our physical Hilbert space,

$$\mathcal{H}_{\text{bulk}} = \left( \bigotimes_{x \in \{x_{\text{bulk}}\}} \mathcal{H}_x \right) \otimes \left( \frac{\bigotimes_{\langle xy \rangle} \mathcal{H}_G^{(xy)}}{\text{Gauss}} \right), \tag{18}$$

where we have labelled each $\mathcal{H}_G$ by the link $(xy)$ it lives on. Now all $i, j$ indices are fixed except for those associated to boundary vertices, and the irrep indices $\mu$ are all related to each other by which fusions are allowed. This completes our description of our new bulk Hilbert space.

Now we turn to the holographic map. We would like a map composed of tensors acting in a spatially local way like in Section 2, but modified to act on this new $\mathcal{H}_{\text{bulk}}$. The trick is we'll first define a new kind of tensor that takes as input a link state in $\mathcal{H}_G$ and outputs a state in a factorized Hilbert space. Specifically, we define a tensor $S_{(xy)} : \mathcal{H}_G^{(xy)} \to \mathcal{H}_G^{xy} \otimes \mathcal{H}_G^{yx}$ to implement the following factorization (the usual embedding into the "extended Hilbert space" [23])

$$S_{(xy)} : \quad |\mu; ij\rangle_G^{(xy)} \longmapsto \frac{1}{\sqrt{d_\mu}} \sum_{k=1}^{d_\mu} |\mu; ik\rangle_G^{xy} |\mu; kj\rangle_G^{yx} . \tag{19}$$

We will draw this as a triple intersection of black lines, so for example

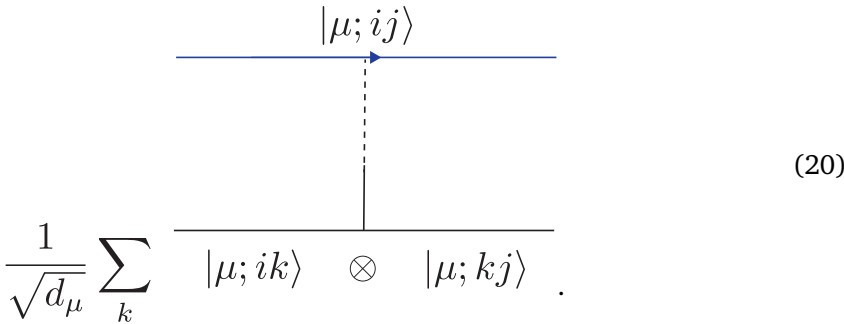

(20)

This should be interpreted as follows. The top (oriented, blue) line represents the state $|\mu; ij\rangle \in \mathcal{H}_G^{(xy)}$. The solid black lines (forming a triple intersection) represent the map (19). The dashed line indicates that the state $|\mu; ij\rangle$ is being input into this map. The output is the state $\sum_k |\mu; ik\rangle |\mu; kj\rangle / \sqrt{d_\mu} \in \mathcal{H}_G^{xy} \otimes \mathcal{H}_G^{yx}$. We'll let

$$S := \bigotimes_{\langle xy \rangle} S_{(xy)}, \tag{21}$$

be the product of all link-factorizers.

We combine this with the old tensors in the following way. After each link has passed through this link-factorizing map, at each vertex we once again have a set of naturally associated Hilbert spaces: the same as (5) but now also with $\mathcal{H}_G$ factors,

$$\mathcal{H}_{T_x'} := \mathcal{H}_x \otimes \left( \bigotimes_{y \text{ nn } x} \mathcal{H}_{xy} \right) \otimes \left( \bigotimes_{y \text{ nn } x} \mathcal{H}_G^{xy} \right). \tag{22}$$

Now as before, for each bulk vertex we pick a state $\langle T_x' |$ in the dual Hilbert space $\mathcal{H}_{T_x'}^*$, giving us a state

$$\bigotimes_{x \in \{x_{\text{bulk}}\}} \langle T_x' | \in \bigotimes_{x \in \{x_{\text{bulk}}\}} \mathcal{H}_{T_x'}^* . \tag{23}$$

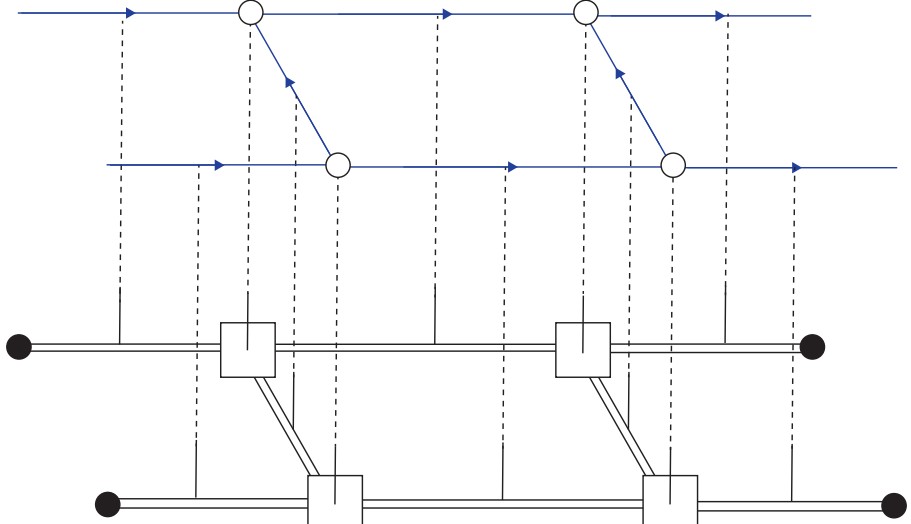

Figure 1: Example tensor network $V$ from (24) acting on $\mathcal{H}_{\text{bulk}}$ from (18). At the top is $\mathcal{H}_{\text{bulk}}$, with the white circles representing the qudits $\mathcal{H}_x$ and the oriented blue lines representing the Hilbert space of the lattice gauge theory $\bigotimes_{\langle xy \rangle} \mathcal{H}_G^{(xy)}/\text{Gauss}$. At the bottom is the tensor network $V$, with the white squares representing one set of tensors (i.e. the states (23)), the triple-intersection black lines representing the link-factorization tensors (19), the black lines (adjacent to them) representing the link state (4), and the black dots representing $\mathcal{H}_{\text{bdry}}$ (or more precisely, the map $W$). The vertical dashed lines indicate where each part of $\mathcal{H}_{\text{bulk}}$ is input into the tensor network.

The boundary vertices we treat by similarly generalizing their $W_x$. The holographic map is then just

$$V = W \left( \bigotimes_{x \in \{x_{\text{bulk}}\}} \langle T_x' | \right) \left( \bigotimes_{\langle xy \rangle} | \phi_{xy} \rangle \right) S. \tag{24}$$

We visualize this acting on (18) as in Figure 1. It is important to note the links of the tensor network are now *double* lined, in contrast to the single lines of (8). This is because every link still carries one line representing the background entanglement (4), but now *also* carries the link-factorization tensor (20). That is,

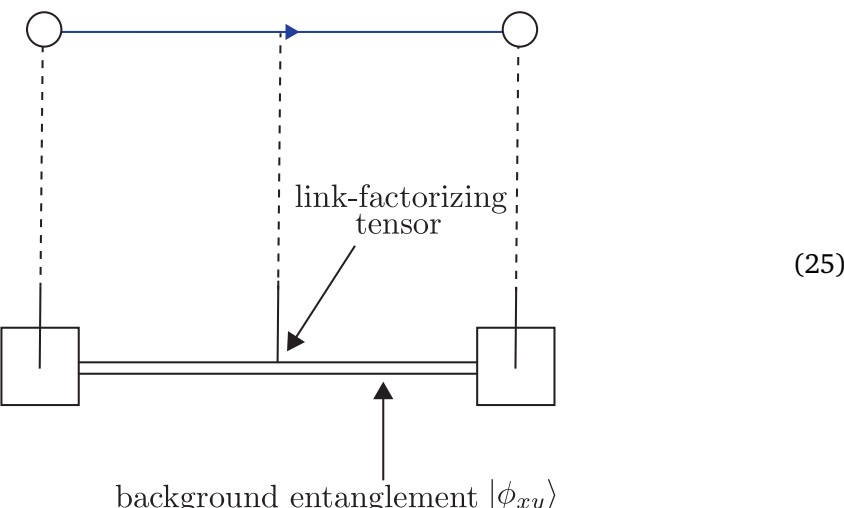

$$\tag{25}$$

Let's now specialize to random tensors for the white boxes. We immediately run into a problem: if $G$ is a Lie group, then $\mathcal{H}_G$ is infinite and therefore so is $\mathcal{H}_{T'_x}$. It is not so clear how to pick a random unitary acting on an infinite dimensional Hilbert space. There are a couple ways to handle this. We will use a simple one, simply truncating each $\mathcal{H}_G$ at some large value of $\mu$, say $\hat{\mu}$ (as in e.g. [17]).

This tensor network now has very nice properties that allow us to compute entropies like before. Indeed, we can obtain a dramatic conceptual simplification by taking all $D_{xy}$ to be very large, much larger than $d_{\hat{\mu}}$. Then nothing stops us from imagining that this is really the original kind of tensor network from Section 2, with $\mathcal{H}_x \to \mathcal{H}_x \otimes \left(\bigotimes_{y \text{ nn } x} \mathcal{H}_G^{xy}\right)$. That is, for $V$ from (24), we can let $V = \widetilde{V}S$ for isometry $\widetilde{V} : S\mathcal{H}_{\text{bulk}} \to \mathcal{H}_{\text{bdry}}$. Given a state $|\psi\rangle \in \mathcal{H}_{\text{bulk}}$, the state $V|\psi\rangle$ is equally well obtained by acting $\widetilde{V}$ on $S|\psi\rangle$. What's nice is that $\widetilde{V}$ is a tensor network just like (7). This helps because we can now import its equations like (12). Let's see how this works. Say we have some state $|\psi\rangle \in \mathcal{H}_{\text{bulk}}$, and we have picked some bulk subregion $b \subseteq \{x_{\text{bulk}}\}$. We want to compute the entropy $S(b)_{S|\psi\rangle}$ of this subregion in the state $S|\psi\rangle$. The relevant density matrix is $\rho$ on $\bigotimes_{x \in b}\left(\mathcal{H}_x \otimes \left(\bigotimes_{y \text{ nn } x} \mathcal{H}_G^{xy}\right)\right)$. Now let $\partial b$ denote the set of links connecting $x \in b$ to $y \in \bar{b}$. We know from $S$ that the $\mu$ in $\mathcal{H}_G^{xy}$ for all $(xy) \in \partial b$ are the same for $\mathcal{H}_G^{yx}$. Therefore those $\mu$ are decohered, and we can decompose $\rho$ to be block diagonal like

$$\rho = \oplus_{\{\mu\}} p_{\{\mu\}} \rho_{\{\mu\}}, \tag{26}$$

where $\{\mu\}$ denotes the set of $\mu$ on the links in $\partial b$ and $p_{\{\mu\}}$ are probabilities. These $\rho_{\{\mu\}}$ are normalized density matrices in which the $\mathcal{H}_G^{xy}$ are restricted to definite values of $\mu$ for $xy \in \partial b$. Importantly, there's even more we can say about these $\rho_{\{\mu\}}$. The form of $S$ tells us that we can write

$$\rho_{\{\mu\}} = \widetilde{\rho}_{\{\mu\}} \otimes \frac{\mathbb{1}_{d_{\{\mu\}}}}{d_{\{\mu\}}}, \tag{27}$$

where $\mathbb{1}_{d_{\{\mu\}}}$ is the identity operator acting on $\bigotimes_{xy \in \partial b} \mathcal{H}_G^{xy}$, and $\widetilde{\rho}_{\{\mu\}}$ is a normalized density matrix acting on the other factors. Therefore,

$$\begin{aligned}
S(b)_{S|\psi\rangle} &= -\operatorname{tr}[\rho \log \rho] \\
&= -\sum_{\{\mu\}} p_{\{\mu\}} \log p_{\{\mu\}} - \sum_{\{\mu\}} p_{\{\mu\}} \operatorname{tr}[\rho_{\{\mu\}} \log \rho_{\{\mu\}}] \\
&= -\sum_{\{\mu\}} p_{\{\mu\}} \log p_{\{\mu\}} - \sum_{\{\mu\}} p_{\{\mu\}} \operatorname{tr}[\widetilde{\rho}_{\{\mu\}} \log \widetilde{\rho}_{\{\mu\}}] + \sum_{\{\mu\}} p_{\{\mu\}} \log d_{\{\mu\}}.
\end{aligned} \tag{28}$$

Now we can use (12) to compute the entropy of boundary subregions:

$$S(B)_{\widetilde{V}S|\psi\rangle} = \min_b \left(\langle\psi|S^\dagger \hat{A}(b) S|\psi\rangle + S(b)_{S|\psi\rangle}\right), \tag{29}$$

with $\langle\psi|S^\dagger \hat{A}(b) S|\psi\rangle = \sum_{xy \in \partial b} \log D_{xy}$ (here for simplicity specializing to the case $|\phi_{xy}\rangle = |\text{MAX}\rangle$). This is an interesting answer, but we would like the answer for $V$ instead of $\widetilde{V}$. Fortunately, by construction $S(B)_{V|\psi\rangle} = S(B)_{\widetilde{V}S|\psi\rangle}$! All that's different is the interpretation of the terms on the right hand side of (29). For $V$, the factors $\mathcal{H}_G^{xy}$ are not part of $\mathcal{H}_{\text{bulk}}$, and so their entropy – the third term in the last line of (28) – is not a part of $S(b)_{|\psi\rangle}$. Instead, those terms must be interpretted as part of the *area* term. We end up with

$$S(B)_{V|\psi\rangle} = \min_b \left(\langle\psi|\hat{A}'(b)|\psi\rangle + S(b)_{|\psi\rangle}\right), \tag{30}$$

where

$$\hat{A}'(b) = \sum_{(xy) \in \partial b} \log(D_{xy})\mathbb{1} + \sum_{\{\mu\}} p_{\{\mu\}} \log(d_{\{\mu\}})\mathbb{1}_{d_{\{\mu\}}}, \tag{31}$$

and $S(b)_{|\psi\rangle}$ is just the first two terms in the last line of (28).

This is a success! The area operator is no longer proportional to the identity. As emphasized in [15–17], this is exactly what we would like for modelling a gauge field on top of a given fixed background. The first term of (31) represents the area from the fixed background, while the second term represents the contribution from the gauge field. This is like the results of [15–17]. Indeed, while the construction of our tensor network (24) differed in some details from [15–17], the main idea is similar pertaining to these two terms. See equations (4.15) and (4.16) of [15], equation (2.26) of [16], and equations (4.11) and (4.12) of [17].

# 4 Background independence

One can wonder: do tensor networks *need* this background entanglement? Could one work if we took it away, $D_{xy} \to 1$? After all, this would be desirable for matching AdS/CFT: in gravity we expect that given a QES, we can project onto states with fairly arbitrary values of the area [24, 25]. In contrast, given area operator (31) there is a minimum value of the area, $\sum_{(xy)\in\partial b} \log D_{xy}$. That's not terrible: it still models the encoding of subspaces of the AdS Hilbert space $\mathcal{H}_{\text{AdS}}$ in which all states have values of the area larger than some minimum value, which happens if we have a fixed background geometry. But it would be nice to have a tensor network without this minimal value, to model the encoding of larger subspaces of $\mathcal{H}_{\text{AdS}}$ without a common background.

At first, it might seem like we can't remove the background entanglement. Indeed, our derivation of (30) used that the $D_{xy}$ were very large. Nevertheless, it turns out that this was just a convenient simplification, and not a necessary part of a good tensor network! The point of this section is to explain carefully how we still get a good holographic tensor network – with a quantum minimal surface formula – even without the background entanglement. The entanglement necessary for the map to be suitably isometric will instead come from the factorization of the link degrees of freedom with (19). Certain states of the link degrees of freedom will be good and holographic.

## 4.1 Background independent tensor network

The tensor network we'll consider will act on the same $\mathcal{H}_{\text{bulk}}$ as before, (18). All that is different is that instead of (24), the tensor network is now

$$V = W \left( \bigotimes_{x \in \{x_{\text{bulk}}\}} \langle T'_x| \right) S, \tag{32}$$

where these $\langle T'_x|$ are defined as in (23) with $D_{xy} = 1$. We depict this graphically in an example in Figure 2.

Let's convince ourselves this works, with an example. Let's say the $\langle T'_x|$ are random tensors, and let's specialize to $G = SU(2)$ for definiteness. As a warmup, consider the following lattice

gauge theory state:

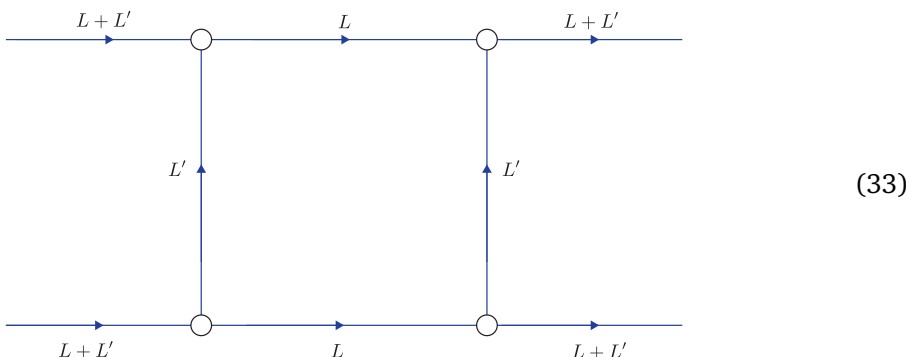

$$(33)$$

We have labelled the value of $\mu$ on each link, here all definite numbers $L, L', L + L'$. We have neglected the $i, j$ indices of $\mathcal{H}_\mu, \mathcal{H}_{\mu^*}$ from (14) because they are unimportant (and again the internal ones are fixed by gauge invariance). We'll imagine the qudits living on the vertices are all together in some state $|\widetilde{\psi}\rangle \in \left(\bigotimes_{x \in \{x_{\text{bulk}}\}} \mathcal{H}_x\right) \otimes \mathcal{H}_R$, where $R$ is some reference system introduced for generality to purify the qudits. Let the total state of the lattice gauge theory, qudits, and reference be $|\psi\rangle$.

Now take this $|\psi\rangle$ and act our tensor network (32). The first step is to apply $S$, which embeds each link into the factorized Hilbert space. Next we apply the random tensors, which for each $x \in \{x_{\text{bulk}}\}$ was

$$\langle T'_x | \in \left(\mathcal{H}_x \otimes \mathcal{H}_G^{xy_1} \otimes \mathcal{H}_G^{xy_2} \otimes \mathcal{H}_G^{xy_3}\right)^* . \tag{34}$$

What's important now is that we have specialized to particular irreps on each link. So for example, at any given vertex we have

$$\langle T'_x | \in \mathcal{H}_x^* \otimes (\mathcal{H}_L \otimes \mathcal{H}_{L^*})^* \otimes (\mathcal{H}_{L'} \otimes \mathcal{H}_{L'^*})^* \otimes (\mathcal{H}_{L+L'} \otimes \mathcal{H}_{(L+L')^*})^* . \tag{35}$$

Recall that $\langle T'_x |$ will act on the state $S |\psi\rangle$. Let's investigate the structure of $S |\psi\rangle$ on these factors. In (35), within each of these parentheses we have two factors. In the state $S |\psi\rangle$, one

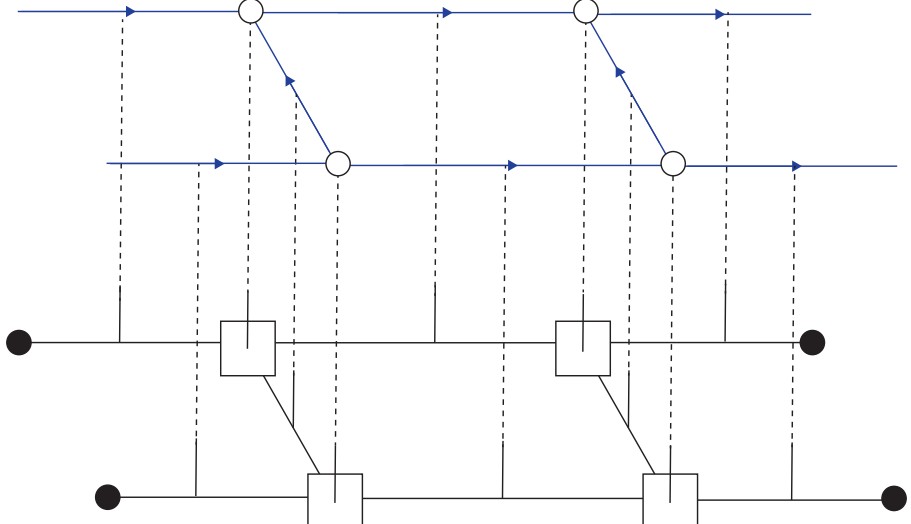

Figure 2: Example tensor network $V$ from (32) acting on $\mathcal{H}_{\text{bulk}}$ from (18). Everything is the same as in Figure 1, except there is no background entanglement, i.e. we have set $D_{xy} = 1$. This is depicted here by each white square tensor being connected by only one line – the tensor (20).

factor from each is fused together by the Clebsch-Gordan coefficients into a definite state, say $|\Omega_{L,L',L+L'}\rangle$. So the only part of $\langle T'_x|$ that matters is the part that looks like $\langle t_x^{L,L',L+L'}|\,\langle\Omega_{L,L',L+L'}|$, where say

$$\begin{aligned}
\langle t_x^{L,L',L+L'}| &\in \mathcal{H}_x^* \otimes \mathcal{H}_{L^*}^* \otimes \mathcal{H}_{L'}^* \otimes \mathcal{H}_{L+L'}^*\,, \\
\langle\Omega_{L,L',L+L'}| &\in \mathcal{H}_L^* \otimes \mathcal{H}_{L'^*}^* \otimes \mathcal{H}_{(L+L')^*}^*\,,
\end{aligned} \tag{36}$$

and $\langle t_x^{L,L',L+L'}|$ is random within that Hilbert space. The part of $\langle T'_x|$ outside the subspace where those three factors are in $\langle\Omega_{L,L',L+L'}|$ just has zero overlap with $S|\psi\rangle$, because $|\psi\rangle$ satisfies Gauss' law. And with extraordinarily high probability over the choice of random unitary, $\langle T'_x|$ will overlap this subspace. (Alternatively, instead of picking $\langle T'_x|$ completely at random, we could just choose to define $\langle T'_x| = \sum_{\mu,\mu',\mu''} \langle t_x^{\mu,\mu',\mu''}|\,\langle\Omega_{\mu,\mu',\mu''}|$ with just $\langle t_x^{\mu,\mu',\mu''}|$ random, similar to [16]. The end result is similar.)

The other factor from each is maximally entangled with factors associated to other vertices, e.g. $\mathcal{H}_G^{xy_1}$ is maximally entangled with $\mathcal{H}_G^{y_1x}$. As a result, this $\langle t_x|$ is exactly like the random tensors from the original tensor network (7), with $\mathcal{H}_{xy}$ replaced by this $\mu$-sector of $\mathcal{H}_G^{xy}$. In other words, this tensor network, acting on this state with fixed irreps, encodes the state of the qudits exactly like a tensor network from Section 2:

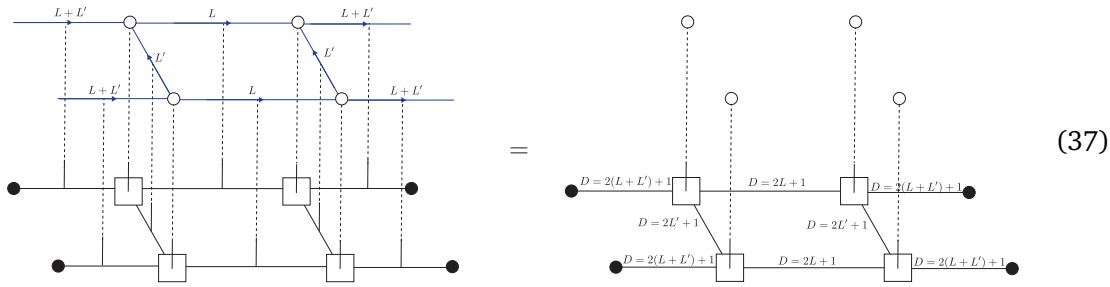

$$\tag{37}$$

This is the main result of this section. Let's say it differently using equations. We recall $S$ maps each link

$$|\mu;ij\rangle_{ab} \longmapsto \frac{1}{\sqrt{d_\mu}}\sum_{k=1}^{d_\mu} |\mu;ik\rangle_{ac}\,|\mu;kj\rangle_{db} = |\mu;ij\rangle_{ab}\left(\frac{1}{\sqrt{d_\mu}}\sum_{k=1}^{d_\mu}|\mu;kk\rangle_{cd}\right), \tag{38}$$

where we have labelled each $\mathcal{H}_\mu, \mathcal{H}_{\mu^*}$ factor by $a,b,c,d$ to help keep track of them. The second factor is a $\mu$-dependent maximally entangled state that was just added by $S$. The first factor is the part of the Hilbert space that was already there in $\mathcal{H}_{\text{bulk}}$, and it remains in the original bulk state. It is the second factor that effectively plays the role of the $|\phi_{xy}\rangle$ of the conventional tensor networks in Section 2. The $\langle t_x^{L,L',L+L'}|$ play the role of the $\langle T_x|$. It follows from all of this that the entropy of boundary subregions satisfies a quantum minimal surface formula,[12] but we will delay discussing it because its nicest features will become clearer.

Given this match (37), what have we gained? The advantage happens when we consider more general lattice gauge theory states. For example, consider the superposition

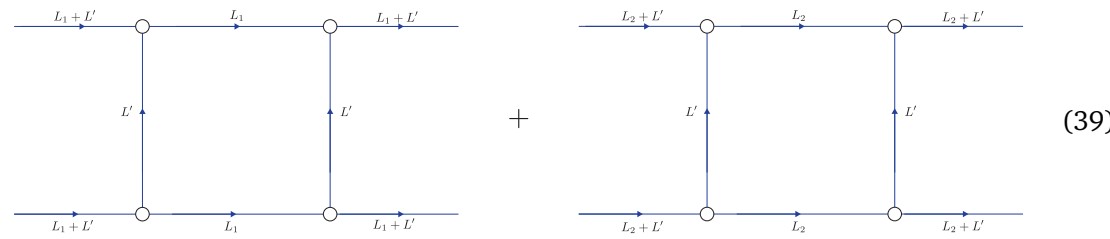

$$\tag{39}$$

---

[12]To get a nice formula with small corrections now requires that $L$ and $L'$ are sufficiently big.

for $L_1 \neq L_2$. Our tensor network (32) acts linearly, and so by (37) the boundary state can be thought of as the superposition of the states prepared by two tensor networks with different background geometries:

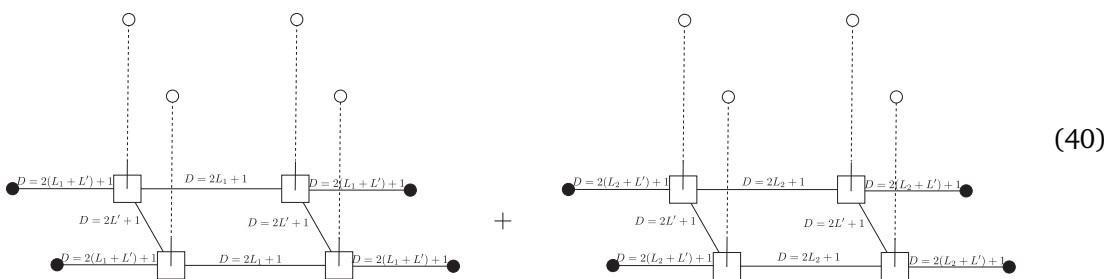

(40)

This lesson generalizes. States in $\mathcal{H}_{\text{bulk}}$ with definite values of $\mu$ on each link can be understood in terms of a conventional tensor network with background entanglement determined by the gauge theory state. Superpositions of these gauge theory states can be understood as superpositions of conventional tensor networks.

Part of this was said before: It has already been said that adding link degrees of freedom can be understood as allowing superpositions of tensor networks, see e.g. [17]. What is new here is that the conventional tensor networks in the superposition need not have any similarities in their geometry. We can make this new feature especially striking: nothing stops us from considering an all-to-all $\mathcal{H}_{\text{bulk}}$, in which all pairs of vertices are connected by a link. Then the "background geometry" of the corresponding conventional tensor networks can look completely different: some states of $\mathcal{H}_{\text{bulk}}$ might have the topology of a (triangulation of a) disk, while others are (triangulations of) higher genus surfaces. This is because the trivial irrep $\mu_0$ is one dimensional, $d_{\mu_0} = 1$, and so a link assigned that irrep has no entanglement across it, hence contributing zero area, and we might as well regard it as not even being there.

It follows from the above that for many gauge theory states (with e.g. sufficiently large $\mu$ on the links), these background independent tensor networks (32) satisfy a quantum minimal surface formula,

$$S(B)_{V|\psi\rangle} = \min_b \left( \langle \psi | \hat{A}(b) | \psi \rangle + S(b)_{|\psi\rangle} \right),$$

(41)

where

$$\hat{A}(b) = \sum_{\{\mu\}} p_{\{\mu\}} \log(d_{\{\mu\}}) \mathbb{1}_{d_{\{\mu\}}},$$

(42)

with $\{\mu\}$ the set of irreps on the links in $\partial b$ (as in (26)) and $S(b)_{|\psi\rangle}$ the first two terms in the last line of (28). This is exactly what we wanted: the area of any given surface is determined completely by the state of the gauge field, with no extra term providing a minimum.

## 4.2  $1d$ **and background independence**

Here's an interesting subtlety. Let's try to use a background independent tensor network like (32) with a 1-dimensional version of $\mathcal{H}_{\text{bulk}}$ from (18), like

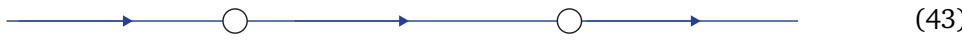

(43)

The problem is that gauge invariant states have exactly the same $\mu$ on each link; that's what Gauss' law says. To see why this is bad, let's again specialize to $G = SU(2)$, where we find that

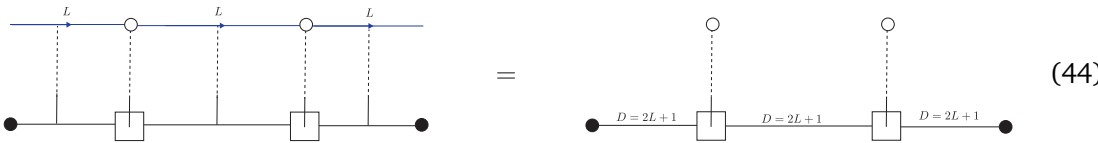

(44)

Now let's try to compute the entropy $S(B)$ for $B$ either of the two black circles. All bond dimensions are equal, so the *minimal b* in the formula (41) can have many problems. For example, it can be maximally degenerate, with every cut an equally minimal surface, say if the qudits are in a product state. More problematically, if any bulk legs are in a mixed state, then neither boundary region will include that leg in its minimal $b$. (Though the entire boundary might still include the entire bulk in its minimal $b$.) This is not like what we expect in 1-dimensional versions of AdS/CFT, where small amounts of bulk entropy are not enough to greatly change the position of the quantum extremal surface of the left or right boundary.

We can improve this model by incorporating charged matter. Here is an example. Let's add to our $\mathcal{H}_{\text{bulk}}$ from (18) an additional degree of freedom at every bulk vertex $x$, with Hilbert space $\mathcal{H}_{x,c} = \oplus_\mu \mathcal{H}_\mu$. Here $\mu$ labels the irreps of $G$, just like in (14). Gauss' law now requires that at every vertex, the two gauge field links *and* this charged matter fuse to the trivial irrep. So now we can have bulk states like

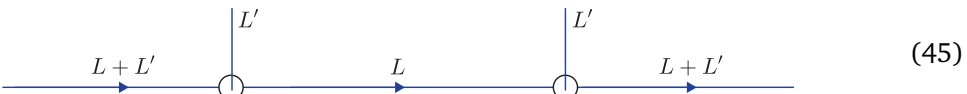

(45)

where again the $L, L'$ label the irreps, and now we have included vertical lines on each bulk vertex to represent the new charged degree of freedom. The (appropriate generalization of the) tensor network (32) now satisfies

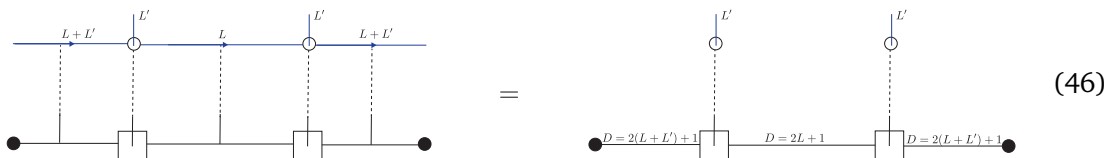

(46)

On the right hand side, the charged matter is labelled $L'$ to remind us that its state is related to the dimension on the bonds. This is better, because the bond dimensions of each link can be different from one another.

## 5  Conclusion

We have shown that there is a straightforward way to build holographic tensor networks that include geometries that are completely distinct, but nonetheless all must satisfy some constraints. This is a step towards adding time evolution in a way that resembles gravity. There, the constraints play a key role in having the bulk dynamics match those of the dual CFT. It will be interesting future work to incorporate different constraints, beyond just Gauss' law, making the bulk theory more like gravity and the holographic map more realistic.

## Acknowledgments

We thank Ronak Soni for encouraging us to finally write this up and useful feedback on the draft. We also thank Juan Maldacena for helpful discussions.

**Funding information**    CA is supported by National Science Foundation under the grant number PHY-2207584, the Sivian Fund, and the Corning Glass Works Foundation Fellowship.

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
