# Peer review of "Background independent tensor networks"

_SciPost Physics, doi:SciPost Phys. 17, 090 (2024)_

## Round 2 · Referee Report · Anonymous (Referee 1) · 2024-7-10

Strengths

  1. The paper is well-motivated, novel, and thought-provoking
  2. To the best of my knowledge, all formulas appear correct
  3. Good discussion of the status of the field, recent progress, and challenges remaining

Weaknesses

  1. There are some places where the presentation was a bit unclear to me

Report

The authors address the topic of designing tensor networks that better model aspects of holographic duality. In particular, they discuss how a central feature of gravity, background independence, can be incorporated into tensor networks that naively appear to have background structure. This is an interesting advance/observation and provides a another demonstration that tensor networks can be pushed even further in modeling AdS/CFT than originally expected.

Requested changes

  1. On page 2, it says "However, it fails to model the fact that in gravity we do not allow arbitrary quantum states on arbitrary geometries. There are constraints that the geometry must satisfy."

It was unclear to me precisely what the criticism was here. In the superposition over geometries, is it not sufficient for the matter to satisfy on the constraints on each geometry separately?

  1. On page 2, there is a type "geomtries"

  2. On page 8, it says "We want something like the one in Section 2..."

I found this vague. What aspects of the tensor network from section 2 do the authors wish to preserve?

  1. In the construction of the tensor network, it is correct that "edge modes" are placed on every link? Is this meaningfully different from only placing edge modes at the "entangling surface" as is usually done in calculations of entanglement entropy in gauge theories?

  2. On Page 12, it says "This is the correct answer..."

What does "correct" mean here? What is this being compared to?

  1. On Page 15, it says "with extraordinarily high probability..."

Does this mean something precise or just colloquial?

  1. For equation (4.10) to be valid, is it necessary for the state to only have support for large $\mu$ everywhere?

  2. On page 18, it says "it can be highly degenerate..."

I do not see an issue with this. Degenerate minimal surfaces have been considered in holography before. What is the problem that the authors are referring to?

  1. At the end of the same paragraph, it says that "this is not like what we expect in 1-dimensional version of AdS/CFT"

How so? It would be helpful if the authors clarified what is the expectation in 1d AdS/CFT.

  1. On page 19, it says "bulk fusion coefficients."

What precisely is meant by this, the bulk qft coupling constants?

  1. I found more generally section 4.3 to be way too quick to get anything meaningful out of if one had not already read the papers they reference [26,27]. I would recommend either the authors briefly review the main points of these papers or remove this section.

Recommendation

Publish (easily meets expectations and criteria for this Journal; among top 50%)

  • validity: top
  • significance: high
  • originality: top
  • clarity: good
  • formatting: excellent
  • grammar: excellent

Author:  Christopher Akers  on 2024-07-25  [id 4654]

(in reply to Report 1 on 2024-07-10)

Thank you for the thorough set of suggestions! We have modified the draft, and list the particular changes below.

  1. We have reworded this sentence to hopefully clarify. We agree that each geometry must satisfy the constraints separately. Our criticism was that in conventional tensor networks, there is no constraint relating the bulk state and the geometry, so even if you take a superposition of two of them, on each branch there is no relationship between the bulk state and the geometry.

  2. Fixed

  3. Good point, we have adjusted this sentence to clarify that we would like a linear map composed of ``tensors’’ acting in a spatially local way.

  4. This is a great question. Yes, this tensor network inserts ``edge modes’’ on every link. It turns out that the entropy of a bulk subregion only receives contributions from the edge modes inserted along the boundary of the subregion; those inserted inside are entangled with other edge modes inserted inside. However, it is vital that we insert them everywhere because we do not know ahead of time which entangling surface we will consider. Said differently, inserting edge modes everywhere is part of the definition of the holographic map that gives us the boundary state. After obtaining the boundary state we can consider different boundary subregions and what their entanglement wedge is — and we are not allowed to choose a different holographic map depending on the choice of boundary subregion we will ask about.

  5. Good point, we have changed the sentence. It now simply emphasizes that this is an interesting answer, but we would like the one for V instead.

  6. This phrase is meant to reference a precise technical result from ``measure concentration’’. The probability that the overlap is smaller than a certain amount epsilon is very increasingly suppressed as the dimension of the Hilbert space grows. This is discussed in some of our references in detail, and we think it might be distracting to delve into it too much in this paper.

  7. (4.10) is cleanest if all links have support on only large mu everywhere, but it is not necessary. For example you can have some links with support only on the trivial irrep, and the holographic map will behave just fine, as though that link is not even there. Issues arise when some tensors only have small mu links adjacent to them, because random states on a small Hilbert space are not as well behaved. The ``measure concentration’’ results place worse bounds.

  8. We agree having degenerate minimal surfaces is not necessarily a problem. The issue is that it is necessarily maximally degenerate — all cuts are equally minimal. This is not necessarily terrible, but it’s not like 1+1d holography. We have modified that sentence to clarify this point.

  9. Thanks, we have added to this sentence to clarify.

10,11. We agree — section 4.3 was probably more confusing to include than it is worth. We have removed it.

---

## Round 3 · Referee Report · Anonymous (Referee 2) · 2024-7-27

Report

The authors have satisfactorily addressed all of my questions/comments.

Recommendation

Publish (easily meets expectations and criteria for this Journal; among top 50%)

---

## Round 3 · Referee Report · Anonymous (Referee 1) · 2024-9-2

Strengths

The paper is clearly written. It has a nice idea of applying properties of lattice gauge theory in the construction of holographic network to make it more graph independent.

Weaknesses

The construction is still at a very primitive state. The space of possible tensors remain extremely large and there are very few guiding principles (which is a general issue confronted by holographic tensor networks) to make progress towards making the bulk geometry concrete.
As it stands, it remains hard to include dynamics which is another long standing issue in tensor network constructions.
There should be in the future more work done in restricting the group G and perhaps restricting or replacing the random tensors.

Report

The paper makes use of graph independence in constructing gauge invariant lattice gauge theory wave-function to construct graph independent holographic tensor network. The construction is reasonable, and it is perhaps as concrete as one could achieve at present from considerations pertaining to entanglement pattern and some level of graph independence.
The authors have also addressed other comments of referee 1.
I would recommend the paper for publication in sci-post.

Recommendation

Publish (easily meets expectations and criteria for this Journal; among top 50%)

---

## Round 3 · List of Changes

We have made a number of changes listed in response to the referee's detailed suggestions.

---

## Editorial Decision

published